# AutoEval Done Right: Using Synthetic Data for Model Evaluation

Pierre Boyeau [1]   Anastasios N. Angelopoulos [1]   Tianle Li [1]   Nir Yosef [1 2]   Jitendra Malik [1]   Michael I. Jordan [1 3]

## Abstract

The evaluation of machine learning models using human-labeled validation data can be expensive and time-consuming. AI-labeled synthetic data can be used to decrease the number of human annotations required for this purpose in a process called *autoevaluation*. We suggest efficient and statistically principled algorithms for this purpose that improve sample efficiency while remaining unbiased.

## 1. Introduction

Our goal is to evaluate machine learning systems—assessing their accuracy, fairness, and other metrics—with as few data points as possible. This goal is important for reducing the human effort required to collect extensive validation datasets (Hastie et al., 2009) for such tasks. Towards that end, we will explore an approach called *autoevaluation*, wherein we evaluate models in a two-stage procedure: (i). Produce synthetic labels using AI on a large unlabeled dataset, and (ii). evaluate AI models using the synthetic labels.

Autoevaluation can save months or years of time and potentially millions of dollars in annotation costs; see, e.g., Scale AI, or the recent work of Zheng et al. (Zheng et al., 2024) using gpt-4 to rank alternative language models' answers to questions with high agreement with human annotators. However, the synthetic labels may not be trustworthy, especially for the purpose of certifying a model's worst-case safety, multi-group accuracy and fairness, or to understand if observed differences between models are significant. This motivates the need for serious statistical inquiry on the general question of autoevaluation.

This work introduces methods for *autoevaluation done right*. Given a small amount of human data and a large amount of synthetic data, we will construct autoevaluation procedures that combine these datasets to get better estimates of performance.

In other words, our methods will increase the effective sample size of human data without compromising statistical validity. Intuitively, we use the limited human data in order to measure the bias of the synthetic data. Then, we evaluate the model on the synthetic data and correct the bias using this estimate. The core statistical tool used for this debiasing is called prediction-powered inference (PPI) (Angelopoulos et al., 2023a); we will describe this tool in detail in the coming text. This approach can improve both metric-based evaluations (Section 2) and pairwise-comparison-based evaluations (Section 3), and can readily be applied using an existing Python software. We will include code snippets throughout for producing more precise unbiased evaluations. These lower-variance evaluations are also accompanied by confidence intervals.

### 1.1. Related Work

Autoevaluation has been a subject of interest, particularly in language modeling, well before the current wave of progress in machine learning (Corston-Oliver et al., 2001; Agarwal et al., 2021; Garg et al., 2022). Since the development of powerful machine learning systems such as gpt-4, the accuracy of the annotations that these systems produce has started to approach that of humans (Zheng et al., 2024; Huang et al., 2024), giving substantial credence to autoevaluation as an alternative to human evaluations (Li et al., 2024a).

The prohibitive cost of human annotation has also encouraged the development of automatic metrics used to evaluate model performance without human aid (Papineni et al., 2002; Lin & Och, 2004), representing a distinct but related approach to autoevaluation. Automatic metrics can be computed on the fly, rely on more data points and are hence less noisy, which can be more informative than human evaluations when the latter are scarce (Wei & Jia, 2021). Standard autoevaluation methods are generally ad hoc, and resulting estimates of model performance can systematically differ from those obtained by human evalua-

[1]Department of Electrical Engineering and Computer Sciences, University of California, Berkeley, USA [2]Department of Systems Immunology, Weizmann Institute of Science, Rehovot, Israel [3]Inria, Ecole Normale Supérieure, Paris, France. Correspondence to: Michael I. Jordan <michael.jordan@berkeley.edu>.

*Proceedings of the 42$^{nd}$ International Conference on Machine Learning*, Vancouver, Canada. PMLR 267, 2025. Copyright 2025 by the author(s).

tion (Garg et al., 2022; van Breugel et al., 2023). In parallel, classical solutions for generating confidence intervals, such as rank-sets (Al Mohamad et al., 2021), cannot take advantage of the AI-generated data. It has been unclear how AI-generated data can be *combined* with human data to improve the quality of evaluations. Towards this end, (Chaganty et al., 2018) produced lower-variance estimates of machine translation performance by combining human preferences with automated metrics via control variates.

For the purpose of model training, a number of strategies have been proposed to combine human-derived ground-truth with synthetic labels, e.g., using pseudo labeling (Lee et al., 2013; Arazo et al., 2020) or consistency regularization (Bachman et al., 2014; Laine & Aila, 2016). Unlike these training methods, our work focuses on the reliable evaluation of already trained models, providing statistical guarantees essential for deployment.

Prediction-powered inference (PPI) is a set of estimators that incorporate predictions from machine learning models (Angelopoulos et al., 2023a) to get lower-variance estimators that remain unbiased. In our case, we employ an optimized variant, PPI++ (Angelopoulos et al., 2023b), in order to estimate metrics using synthetic data. From a statistical perspective, PPI is closely related to the fields of multiple imputation and semiparametric inference, perhaps most notably the augmented inverse propensity weighting (AIPW) estimator (Robins & Rotnitzky, 1995; Tsiatis, 2006) (see (Angelopoulos et al., 2023b) for a careful review). Indeed, we are not the first to notice this application of PPI; the work of Saad-Falcon et al. (Saad-Falcon et al., 2023) describes an autoevaluation method for evaluating and ranking language models from pairwise comparisons for the purpose of retrieval-augmented generation. A preprint by Chatzi et al. (Chatzi et al., 2024), posted concurrently with ours, also considers the problem of ranking models from pairwise comparisons, and constructs approximate rankings with coverage guarantees. Our approach is complementary to these existing works. Our specific contribution is to develop an instantiation of PPI that is practical and yields tight confidence intervals, is easy to implement using existing software, and is compatible with existing evaluation systems such as Chatbot Arena (Cha; Chiang et al., 2024). Moreover, we evaluate our PPI method on real data. Along the way, we develop an interesting extension of the PPI algorithms to the case where the annotation model outputs not just a single synthetic $Y$, but a distribution over $Y$.

## 2. Autoevaluating Accuracy and other Metrics

We begin by describing how to use prediction-powered inference for estimating metrics. The most commonly used metrics are accuracy and loss, so we focus on these; however, our tools will be general and allow autoevaluation of any metric.

### 2.1. Defining the Goal

**Basic notation** We observe inputs $X$ in some space $\mathcal{X}$, such as the space of natural images, natural language, and so on. We seek to predict labels $Y$ in some space $\mathcal{Y}$, such as the space of classes, next tokens, actions, etc. Towards this end, let $f_1, \ldots, f_M$ denote $M$ pretrained models mapping inputs in $\mathcal{X}$ to label estimates in some third space $\widehat{\mathcal{Y}}$. We often have $\widehat{\mathcal{Y}} = \mathcal{Y}$, in which case the model directly outputs predictions of the label. However, we leave open the possibility that $\widehat{\mathcal{Y}}$ is some other space—such as the space of softmax scores in the case of classification. The appropriate output space for $f(X)$ will be easy to infer from context.

**Metrics** We will evaluate the performance of the models by estimating the expectation of some metric function $\phi :$ $\widehat{\mathcal{Y}} \times \mathcal{Y} \to \mathbb{R}$; in other words, the *metric* of model $m$ will be

$$\mu_m = \mathbb{E}\left[\phi(f_m(X), Y)\right] \tag{1}$$

for some metric function $\phi$ and every $m = 1, \ldots, M$. We are interested in estimating the $M$-length vector $\mu = (\mu_1, \ldots \mu_M)$. For example, in the case of the accuracy, we would want to measure $\mathsf{accuracy}_m = \mathbb{E}\left[\phi_{acc}(f_m(X), Y)\right]$ where $\phi_{acc}(y, y') = 1$ if $y = y'$ and 0 otherwise, for every $m \in 1, \ldots, M$.

Accuracy is not the only quantity that can be framed within this setup. As another example, when the predictors are multilabel classifiers, one performance metric of interest could be the average precision of the model, that is, $\phi_{AP}(\hat{y}, y) := \frac{|\hat{y} \cap y|}{|\hat{y}|}$ . In the case of regression, $\mu_m$ could correspond to the mean squared or absolute error of model $m$, in which case $\phi(\hat{y}, y) := (\hat{y} - y)^2$ or $\phi(\hat{y}, y) := |\hat{y} - y|$, respectively. Finally, one can imagine estimating multiple losses at once; for example, for the purpose of assessing fairness, one may want to evaluate accuracy across many groups.

**Data** We assume access to two datasets: a small human-annotated dataset, $\{(X_i, Y_i)\}_{i=1}^{n}$, and a large amount of unlabeled data points, $\{X_i^u\}_{i=1}^{N}$, whose ground-truth labels $\{Y_i^u\}_{i=1}^{N}$ are unavailable. Importantly, both datasets are i.i.d.; extensions to some limited non-i.i.d. regimes are handled in (Angelopoulos et al., 2023a), but we will not discuss them here. One should think of the regime where $N \gg n$: we have far more synthetic labels than real ones. For both datasets and every model, we also assume access to a *syn-*

*thetic label distribution* that approximates $p(Y \mid X)$. We denote $\{\tilde{P}_{i,m}\}_{i=1}^{n}$ and $\{\tilde{P}_{i,m}^{u}\}_{i=1}^{N}$ as the set of synthetic label distributions conditioned on the labeled and unlabeled input data points, respectively. For each $i$ and $m$, we will use the notation $d\tilde{P}_{i,m}(y)$ to represent the estimated PDF or PMF evaluated at label $y$.

For the sake of intuition, we make a few remarks regarding this data generating process. First, the synthetic data distributions can be seen as distributions over labels produced by one or several "annotator models", that can either be related or different from the models to evaluate. In the latter case, the synthetic label distribution, for a given input, is the same for each model $f_1, \ldots, f_M$. We do not need the subscript $m$ in this scenario, and can simply denote the synthetic label distribution as $\tilde{P}_i$. However, the general case of $\tilde{P}_{i,m}$ allows for each model to have a different annotator model, and possibly allow models to self-annotate, that is, to themselves produce synthetic labels. Second, we note that the framework we described applies directly to the case where the annotator model produces single predictions of $Y$ instead of distributions, by setting up $d\tilde{P}_{i,m}(y)$ to be a delta function at the prediction (the distribution is entirely concentrated on the prediction of $Y$).

## 2.2. The Algorithm

We combine the labeled and unlabeled data to estimate $\mu$. In particular, we seek to benefit from the large sample size of the automatically annotated dataset to produce an estimator with low variance, while ensuring that this estimator remains unbiased. We will begin with the case of estimating accuracy, and then generalize our algorithm to arbitrary metrics.

### WARM-UP: MODEL ACCURACY

The classical approach to estimating model accuracy is to compute the fraction of correct labels:

$$\hat{\mu}_m^{\text{classical}} = \frac{1}{n} \sum_{i=1}^{n} \mathbb{1}(\hat{Y}_{i,m} = Y_i),$$

where $\hat{Y}_{i,m} = \arg\max_y f_m(X_i)_y$ and $f_m(X_i)$ is the softmax output of model $m$. Instead, we propose estimating the accuracy of a classifier differently: by using the classifier's own confidence on the unlabeled data as a signal of its accuracy. Let $p_{i,m} = f_m(X_i)_{\hat{Y}_{i,m}}$ denote the top softmax score of model $m$ on labeled example $i$, and $p_{i,m}^u, \hat{Y}_{i,m}^u$ be defined analogously. We will use the estimator

$$\hat{\mu}_m := \underbrace{\frac{\lambda}{N} \sum_{i=1}^{N} p_{i,m}^u}_{\text{accuracy}} + \underbrace{\frac{1}{n} \sum_{i=1}^{n} \Delta_{i,m}^{\lambda}}_{\text{bias correction}}, \qquad (2)$$

where $\Delta_{i,m}^{\lambda} := \mathbb{1}\{\hat{Y}_{i,m} = Y_i\} - \lambda p_{i,m}$. Here, $\lambda$ is a tuning parameter—for the time being, think of $\lambda = 1$. The above estimator decomposes into two natural components. Interpreting the top softmax score as the probability of correctness, the first term captures the model's internal estimate of its accuracy on the unlabeled data. The second term is the bias of the first term.

This estimator has two beneficial properties: *unbiasedness* and *variance reduction*. Unbiasedness means that $\mathbb{E}[\hat{\mu}] = \mu$. This implies that the inclusion of machine learning predictions in our estimator does not introduce systematic errors for estimating the accuracy. Variance reduction means that the use of synthetic data reduces the variance of our estimator: $\text{Var}(\hat{\mu}_m) \leq \text{Var}(\hat{\mu}_m^{\text{classical}})$.

This is formally true for the optimally chosen parameter $\lambda$; indeed, the optimal choice of $\lambda$ ensures that our estimator is always better than $\hat{\mu}^{\text{classical}}$ (in an asymptotic sense). See (Angelopoulos et al., 2023b) for details and a formal proof; refer to Supplement A to see how to compute this estimator in Python.

### GENERAL METRICS

The approach we have presented for evaluating classifier accuracy is an instance of a more general framework for evaluating properties of machine learning models. In particular, we can use our annotator model to output an approximate expectation of each label $\{Y_i\}_{i=1}^{n}$ and $\{Y_i^u\}_{i=1}^{N}$ in the following way:

$$\begin{cases} \hat{\mathbb{E}}_{i,m} &= \int_{y \in \mathcal{Y}} \phi(f_m(X_i), y) d\tilde{P}_i(y) \\ \hat{\mathbb{E}}_{i,m}^u &= \int_{y \in \mathcal{Y}} \phi(f_m(X_i^u), y) d\tilde{P}_i^u(y). \end{cases} \qquad (3)$$

These expressions look complicated, but have a simple interpretation: the annotator model, given $X_i$, thinks the distribution of $Y_i$ is $d\tilde{P}_i$, and we are simply calculating the expected metric under that estimated distribution. This explains the hats on the expectation symbols; these are not real expectations, but rather, estimated expectations according to the annotator model. Indeed, in the case of classification, we see that, as is intuitive, the expected accuracy of the $m$th model on the $i$th data point is equal to its top softmax score:

$$\hat{\mathbb{E}}_{i,m} = \int \phi_{acc}(\hat{Y}_{i,m}, y) d\tilde{P}_{i,m}(y) = d\tilde{P}_{i,m}(\hat{Y}_{i,m}) = p_{i,m}.$$

Along the same lines, our previous estimator can be generalized to the case of arbitrary metrics as

$$\hat{\mu}_m := \underbrace{\frac{\lambda}{N} \sum_{i=1}^{N} \hat{\mathbb{E}}_{i,m}^u}_{\text{metric on synthetic data}} + \underbrace{\frac{1}{n} \sum_{i=1}^{n} \Delta_{i,m}^{\lambda}}_{\text{bias correction}}, \qquad (4)$$

where now $\Delta_{i,m}^{\lambda} := \phi(f_m(X_i), Y_i) - \lambda\hat{\mathbb{E}}_{i,m}$. The first sum in the above expression is the average metric predicted by model $m$ over all synthetic labels. If the annotator model is near-perfect and $N$ is large, then this term will almost exactly recover the metric. However, if the synthetic label distribution is not good, this can bias our estimate of the metric. The second term corrects this bias by calculating it on the labeled dataset and subtracting it off.

Returning to the role of the tuning parameter: $\lambda \in [0, 1]$ is a discount factor on our synthetic data. When the synthetic data is very good, we can set $\lambda = 1$; when it is bad, setting $\lambda = 0$ will throw it away entirely. One can asymptotically optimize the variance of $\hat{\mu}$ in order to set $\lambda$, as in (Angelopoulos et al., 2023b).

Again, it is straightforward to see that for any fixed value of $\lambda$, our estimator in (4) is unbiased, meaning $\mathbb{E}[\hat{\mu}] = \mu$, and will be strictly lower-variance than its classical counterpart when $\lambda$ is optimally chosen.

VARIANCE AND CONFIDENCE INTERVALS

As we have explained above, the main benefit of AutoEval is to reduce the number of human-labeled data points to achieve a particular variance. We can formalize this by analyzing the variance of $\hat{\mu}_m$ and $\hat{\mu}_m^{\text{classical}}$. In particular, we can write the covariance matrix of $\hat{\mu}$ as

$$\frac{1}{n}V = \frac{1}{N}\lambda^2\text{Cov}(L_i^u) + \frac{1}{n}\text{Cov}(\Delta_i^{\lambda}),$$

where $\Delta_i^{\lambda} := (\Delta_{i,1}^{\lambda}, \ldots, \Delta_{i,M}^{\lambda})$. This expression admits a simple plug-in estimator; it also indicates that we should pick $\lambda$ to minimize $V$ in the appropriate sense. It also allows for the production of non-asymptotic confidence intervals using concentration. We opt to use asymptotic confidence intervals for $\mu$. In particular, we have that as $n$ and $N$ approach infinity,

$$\sqrt{n}\widehat{V}^{-1/2}(\hat{\mu} - \mu) \rightarrow \mathcal{N}(0, \mathbb{I}_M),$$

where $\widehat{V}$ is the plug-in estimator of $V$, corresponding to

$$\widehat{V} = \frac{n\lambda^2}{N}\widehat{\text{Cov}}(L_i^u) + \widehat{\text{Cov}}(\Delta_i), \quad L_i^u = (\hat{\mathbb{E}}_{i,1}^u, \ldots, \hat{\mathbb{E}}_{i,M}^u).$$

Note that when $\lambda = 0$, we exactly recover $\hat{\mu}^{\text{classical}}$—but this may not be the parameter that minimizes the variance $\widehat{V}$. Indeed, we can explicitly choose $\lambda$ to minimize the variance. An explicit expression for this estimate can be found in (Angelopoulos et al., 2023b).

Another beneficial aspect of the asymptotic analysis is that it allows us to construct confidence intervals with which we can reliably rank models. For example, coordinatewise, the following is an asymptotically valid $1 - \alpha$ confidence interval *marginally* for each $\hat{\mu}_m$:

$$\mathcal{C}_m = \left(\hat{\mu}_m \pm \frac{z_{1-\alpha/2}}{\sqrt{n}}\widehat{V}_{m,m}\right). \tag{5}$$

The above interval comes with the following (standard) guarantee for all $m = 1, \ldots, M$: $\lim_{n,N\to\infty} \mathbb{P}(\mu_m \in \mathcal{C}_m) = 1 - \alpha$.

As an alternative to producing confidence intervals for a single coordinate $\mu_m$ based on Equation (5), we might want to create confidence sets that contains the entire vector $\mu$, that is, simultaneously valid intervals. The simultaneous interval can be constructed using the chi-squared distribution as

$$\mathcal{C}^{\chi} = \left\{\mu : n\left\|\widehat{V}^{-1/2}(\hat{\mu} - \mu)\right\|_2^2 \leq \chi_{1-\alpha,M}^2\right\},$$

where $\chi_{1-\alpha,M}^2$ denotes the $1-\alpha$ quantile of the chi-squared distribution with $M$ degrees of freedom. This interval has the following (standard) guarantee:

$$\lim_{n,N\to\infty} \mathbb{P}(\mu \in \mathcal{C}^{\chi}) = 1 - \alpha,$$

and thus, it can be used to rank the models by checking whether the $m$ and $m'$ coordinates of $\mathcal{C}^{\chi}$ overlap for each model $m$ and $m'$ in $1, \ldots, M$.

**2.3. Application to Rank Computer Vision Models**

We applied the described methodology applies for evaluating computer vision models. We considered five trained computer vision models (ResNet-18, ResNet-34, ResNet-50, ResNet-101, and ResNet-152) optimized over the training set of ImageNet and sourced from PyTorch (Paszke et al., 2019). We considered the task of estimating their accuracy on the validation set of ImageNet in a low-data regime, using a subset of labeled data points. The ground-truth model accuracies were computed as the mean accuracies evaluated over the entire validation dataset.

We considered two different approaches to estimate the accuracy of these models. The first is referred to as PPI (Angelopoulos et al., 2023a), and corresponds to (2) with $\lambda = 1$. The second strategy, PPI++ (Angelopoulos et al., 2023b) optimizes $\lambda$ to minimize the variance, with limited computational overhead (Table S3). These approaches were benchmarked against $\hat{\mu}^{\text{classical}}$ along with a standard z-test confidence interval.

To reflect a low-data regime, we randomly sampled a small number $n$ of observations to be used as labeled data points available for these approaches. The rest of the observations in the validation data were used as unlabeled data points for PPI and PPI++. Our synthetic label distribution $d\tilde{P}_{i,m}$

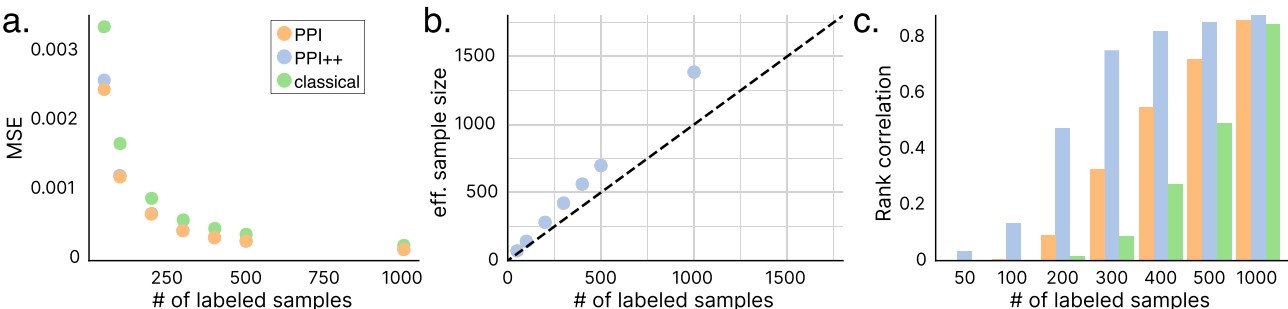

*Figure 1.* **ImageNet experiment.** For every approach, we built confidence intervals around the average accuracy of different ResNet architectures. **a.** MSE of the point estimates of the model accuracies. **b.** ESS of *PPI* and *PPI++* against the classical approach. **c.** Correlation between the estimated and true model rankings. Here, and in all following figures, obtained metrics are averaged across 250 random splits of the validation data into labeled and unlabeled data.

is the softmax vector of model $m$ on labeled data point $i$; $d\tilde{P}_{i,m}^u$ for an unlabeled data point is analogous.

The mean-squared error of our estimates of the model accuracies improved over the classical baseline (Figure 1a). Both PPI and PPI++ had lower mean-squared errors than the baseline, no matter the size of the labeled set. Little to no difference was observed between PPI and PPI++, which probably means that the imputed accuracy scores are reliable proxies for the true quantities. Our approach hence provided more accurate point estimates of the model accuracies. When uncertainty quantification does matter, PPI and PPI++ provided calibrated confidence intervals across all labeled set sizes, and produced tighter confidence intervals than the baseline (Figure 6).

The benefit of using unlabeled data can be measured by computing the effective sample size (ESS) of PPI and PPI++ relative to the classical approach (Figure 1b). This value can be interpreted as the equivalent number of labeled data points for the classical approach that would be required to achieve the same level of precision as PPI or PPI++. Our ESS exceeds that of the classical approach by approximately 50%, which demonstrates the utility of unlabeled data for evaluating model performance.

Here, and in the other experiments, we also evaluated our approach for the purpose of model ranking, by ranking models based on their confidence intervals after Bonferroni correction. Models with overlapping confidence intervals were considered tied. Figure 1c shows the correlation of the estimated model ranks with the ground truth ranking (computed on all data) for different $n$ and averaged across labeled-unlabeled data splits. This experiment showed dramatic differences between the approaches. PPI++ showed much stronger correlations with the ground truth than the other approaches, meaning that its rankings were more accurate and less prone to ties.

To confirm the applicability of our approach to scenarios

using larger sample sizes, we rerun this experiment with $n = 10,000$ labeled data points (Table S4). This experiment confirmed, among other things, that PPI++ provided more accurate point estimates and tighter confidence intervals than the classical approach.

### 2.4. Application to Evaluate Protein Fitness Prediction Models

We also used AutoEval to rank regression models, and more specifically, protein fitness prediction models. Protein fitness prediction is a crucial task in computational biology, aiming to predict the biological relevance of protein mutations based on their amino acid sequences. The recent development of deep learning models for protein language modeling has enabled the emergence of powerful models, trained on millions of protein sequences, used to predict protein fitness in a zero-shot manner (Meier et al., 2021). Unfortunately, evaluating these models for a specific task remains challenging due to the scarcity of experimental data that can be used for evaluation, typically requiring expensive, time-consuming, and poorly scalable wet-lab experiments (Laine et al., 2019; Riesselman et al., 2018).

We applied AutoEval on ProteinGym (Notin et al., 2023), which gathers several assays containing both experimental fitness measurements, used as ground-truth labels, and predicted fitness scores from various fitness predictive models. We focused on ranking protein language models for predicting the fitness of mutations in the IgG-binding domain mutations of protein G based on an assay of $N = 536,962$ pairwise mutations (Olson et al., 2014).

We considered a scenario where one aims to select the best model for zero-shot fitness prediction for a specific protein, using a small experimental dataset and a large set of potential mutations for which fitness is not measured. We focused on the Pearson correlation between predicted and experimental fitness scores as a validation metric for rank-

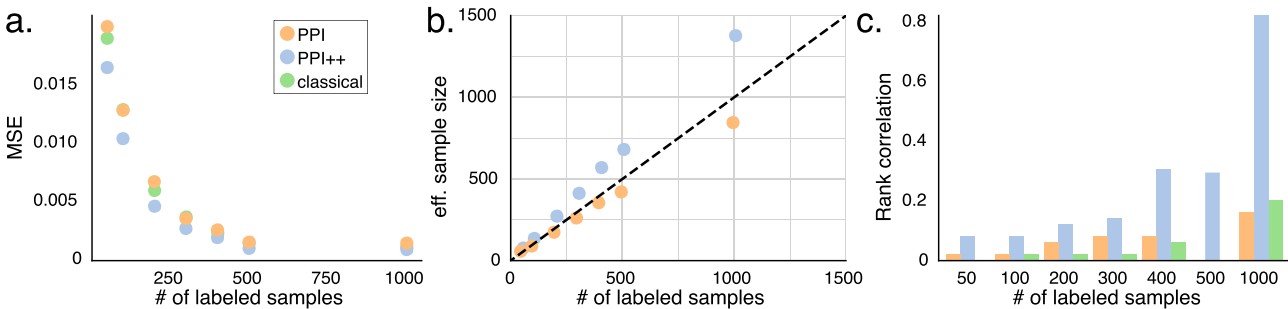

*Figure 2.* **Protein fitness experiment** for building confidence intervals and point estimates for the Pearson correlation of seven protein language models with the experimental fitness scores, using a held-out model to produce synthetic labels. **a.** MSE of the point estimates of the model correlations. **b.** ESS of *PPI* and *PPI++* against the classical approach. **c.** Correlation between the estimated and true model rankings.

ing models. More specifically, we aimed to estimate the metric $\mu_m = \mathbb{E}[Y f_m(X)]$, where $Y, X, f_m$ are the experimental fitness, the protein sequence, and the m-th fitness predictor, respectively, assuming that $Y$ and $f_m(X)$ have zero mean and unit variance. This fits in our general metric evaluation framework, where the metric function in Equation (1) is $\phi(y, y') = yy'$.

We used VESPA (Marquet et al., 2022), a protein language model, as a held-out annotator model for synthetic label generation. VESPA produce point estimates of the experimental fitness scores $f_{\text{VESPA}}(X_i)$ based on the protein sequence $X_i$ aiming to approximate the experimental fitness scores $Y_i$. While VESPA does not provide uncertainty estimates for its predictions, we can still use it as an annotator model in our framework. Specifically, we applied our estimator to estimate model $m$'s Pearson correlation with the experimental fitness scores, by setting $\mathbb{E}_{i,m}^u = f_m(X_i^u) f_{\text{VESPA}}(X_i^u)$ for the synthetic term and $\Delta_{i,m}^\lambda = f_m(X_i) Y_i - \lambda f_m(X_i) f_{\text{VESPA}}(X_i)$ for the bias correction term in Equation (4).

The results of this experiment are shown in Figure 2. The effective sample sizes of PPI++ were systematically higher than the classical approach (Figure 2b), by approximately 50%. Furthermore, the ranks obtained by our approach were also much closer to the true model ranks than the classical approach (Figure 2c), with a five-fold improvement for $n = 1000$.

PPI++ confidence intervals for models' correlations with the experimental fitness scores were also slightly tighter than the classical approach, yet remained calibrated (Figure 6). The PPI estimator performed worse than the classical approach. This is a known issue of this estimator, that PPI++ mitigates.

A question remains: how good does the annotator model need to be to allow AutoEval to work well? Figure 3 compares the effective sample size of PPI++ obtained

with different annotator models. As expected, the better the annotator model, the higher the effective sample size. We importantly note that even with a very poor annotator model, PPI++ performs at least as well as the classical approach. When the annotator labels do not correlate with the true labels, PPI++ falls back to the classical approach ($\lambda = 0$), effectively ignoring the synthetic labels. That being said, we observe that even mediocre annotator models, such as CARP, provide a 10% increase in effective sample size compared to the classical approach. Altogether, these observations suggest that AutoEval can provide better point estimates and tighter confidence intervals compared to the classical approach even when the annotator model is mediocre.

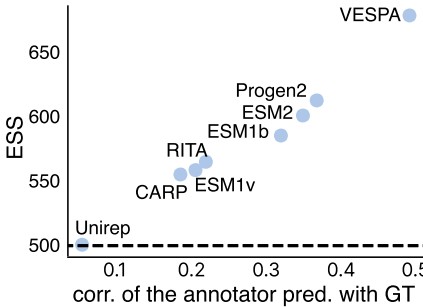

*Figure 3.* ESS of PPI++ against annotator model performance for $n = 500$ labeled points for the protein fitness experiment. The horizontal line denotes the ESS of classical.

## 3. Evaluating Model Performance from Pairwise Comparisons

Characterizing the absolute performance of ML models for the purpose of ranking them is challenging. The previous section described a methodology to compare models based on a common performance metric. Unfortunately, metrics serving as proxies for model performance might either not

exist, or diverge from human judgment (Ji et al., 2023).

In such cases, assessing *relative* model performance might be more appropriate. This can typically be done by comparing different model predictions to each other. The Chatbot Arena project (Cha), for instance, allows human annotators to state preferences over different LLM predictions to the same prompt. Comparison-based evaluation is also an exciting opportunity for autoevaluation (Zheng et al., 2024; Li et al., 2024b). In particular, an external LLM, prompted to serve as an annotator, agrees with human annotators with high fidelity. Still, it is unclear how biased an AI annotator might be, which drastically limits the usefulness of the validation data it produces. (Li et al., 2024b) conducted studies into biases within AI annotators, such as stylistic and model-specific biases, highlighting the need for more robust inference. This section describes how to leverage such AI-generated preferences while making statistically valid inferences about model performance.

### 3.1. A Model to Assess Relative Performance

The canonical model for assessing relative performance of models based on pairwise comparisons, as in a tournament, is called the Bradley-Terry (BT) model (Zermelo, 1929; Bradley & Terry, 1952; Ford Jr, 1957). The BT model is used in the Chatbot Arena (Chiang et al., 2024), by the World Chess Federation, the European Go Federation, and many other competitive organizations as a tool for ranking players.

Now we describe the BT model. Imagine, among $M$ models, we are trying to compare the strength of model $A$ to the strength of model $B$. Towards this end, we give a prompt $Q$ to both models, and they give us an answer. We show this answer to a human, who gives us $Y = 1$ if the answer of model $B$ is better than the answer of model $A$, and vice versa. The assumption of the BT model is that $Y$ follows a logistic relationship,

$$P_\zeta(Y = 1 \mid A, B) = \frac{1}{1 + e^{\zeta_A - \zeta_B}},$$

with some parameter vector $\zeta$ of length $M$, whose entries are called the *Bradley-Terry coefficients*. Each model $m$ has a BT coefficient $\zeta_m$ which, when large relative to the other coefficients, signifies that it is more likely to win the pairwise comparison. (Also, because the model in (3.1) is invariant to addition of a constant to every coordinate of $\zeta$, we can, without loss of generality, set $\zeta_1 = 0$, making the model identifiable.)

It is well-known that, given a labeled dataset of $n$ pairwise comparisons, $\{A_i, B_i, Q_i, Y_i\}$, the maximum-likelihood estimator of the BT coefficients is a logistic regression (Hunter, 2004). Let $X_i$ be the vector of all zeros except at indexes $A_i$ and $B_i$, where it is $-1$ and $1$ respectively. The

logistic regression estimate of the BT coefficients is

$$\hat\zeta^{\text{classical}} = \operatorname*{argmin}_{\zeta \in \mathbb{R}^{M-1}, \zeta_1 = 0} \frac{1}{n} \sum_{i=1}^{n} \ell_\zeta(X_i, Y_i),$$

where $\ell$ is the binary cross-entropy loss.

### 3.2. Autoevaluation of Relative Performance

Prediction-powered inference can be applied out-of-the-box to the BT model, making it possible to leverage large numbers of AI-generated preferences while controlling for their potential bias. In addition to the set of human preferences defined above, additionally define the unlabeled dataset $\{(A_i^u, B_i^u, Q_i^u)\}_{i=1}^{N}$. On both the labeled and unlabeled datasets, we have the prompt and both models' answers; we use a LLM in place of the human to choose between the answers. This gives us a prediction $\hat Y_i$ and $\hat Y_i^u$ of the pairwise comparison on both datasets. The `PPI++` estimator of the BT coefficients is given by

$$\begin{aligned}
\hat\zeta = \operatorname*{argmin}_{\substack{\zeta \in \mathbb{R}^{M-1} \\ \zeta_1 = 0}} &\frac{1}{n} \sum_{i=1}^{n} \left( \ell_\zeta(X_i, Y_i) - \lambda \ell_\zeta(X_i, \hat Y_i) \right) \\
&+ \frac{\lambda}{N} \sum_{i=1}^{N} \ell_\zeta(X_i^u, \hat Y_i^u),
\end{aligned} \tag{6}$$

where $\lambda \in [0, 1]$ controls the weight we give to the AI-generated preferences. Although this estimator departs from the arguments given in Section 2, it has a very similar interpretation; it constructs an unbiased and lower-variance loss function for the true logistic regression, and then minimizes it.

The resulting BT coefficient estimates have the same appealing properties as above. In particular, they are unbiased for any fixed $\lambda$, and one can construct valid confidence intervals around them using `PPI` and `PPI++`; see (Angelopoulos et al., 2023a;b) for this and other generalized linear models, as well as methods for optimally choosing $\lambda$. Supplement A describes how to compute this estimator easily in Python.

### 3.3. Autoevaluation of LLMs from Pairwise Preferences

We evaluated our approach on the Chatbot Arena project (Chiang et al., 2024). We first extracted 16K observations from the Chatbot Arena dataset, in which a total of 20 recent LLMs were compared (Table S1). Each observation contains a prompt written by a human, responses from two of the 20 LLMs, and the preference of the human over the two responses. For each observation, we used `gpt-4o-mini` as a judge (Zheng et al., 2024) by prompting it to identify the most useful response, following the

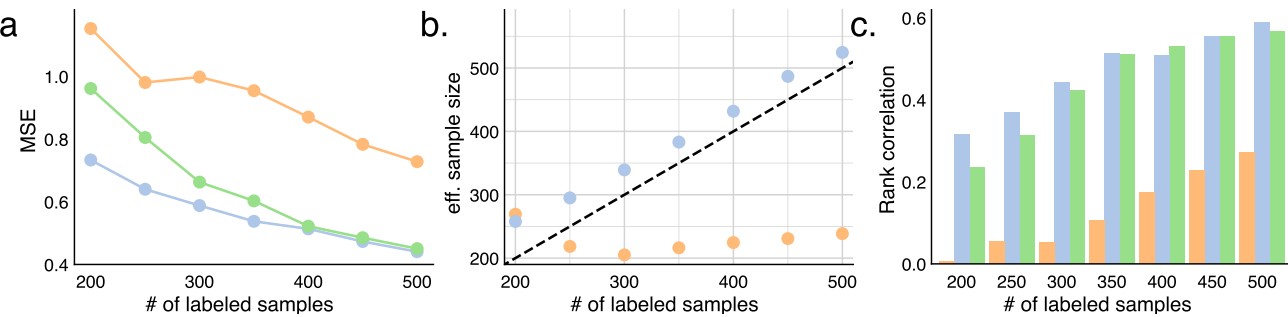

*Figure 4.* **LLM experiment** for building confidence intervals and point estimates for the BT coefficients of different LLMs. **a.** MSE of the point estimates of the BT coefficients. **b.** ESS of PPI and `PPI++` against the classical approach. **c.** Correlation between the estimated and true model rankings.

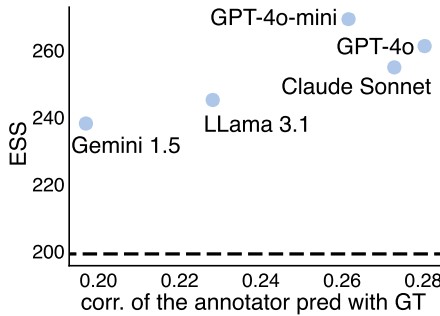

*Figure 5.* ESS of `PPI++` against annotator model performance for $n = 200$ labeled points in the LLM experiment. The horizontal line denotes the ESS of classical.

## Discussion

AutoEval is a promising direction to reduce the cost and effort of model evaluation. We have presented a methodology that makes it possible to use such synthetic data for model evaluation, improving over classical approaches in a statistically rigorous way. Our implementation is available as a Python package.

It is worth noting that the statistical methods we have presented apply beyond autoevaluation. Indeed, many setups involve trading off low-quality or unreliable validation labels, which are plentiful, with high-quality but scarce validation labels. Our methodology applies readily to such setups, and could, for instance, help integrate crowd-sourced validation labels with expert validations.

One limitation of our approach is that it requires the curated expert and autoevaluated data to be representative of the data in production. These distributional shifts typically arise when the labeled inputs are not sampled uniformly at random from the unlabeled pool, or when labeled and unlabeled data points come from different populations altogether. In such cases, AutoEval, in the form described in Equations (4) and (6), loses the statistical guarantees we outline in the paper, and confidence intervals may no longer be valid. To address this issue, we derived alternative AutoEval estimators that are robust to covariate shifts (see Supplement B).

Finally, it is interesting to consider other metrics by which one could evaluate models with `PPI`-type approaches. Herein, we handled mean estimation and logistic regression, but the framework can do more. One might want to evaluate other metrics, such as fairness and bias, e.g., via estimating the least-squares coefficients relating sensitive attributes and prediction error. For any deployed machine learning system, it is important to test many of these metrics to ensure good performance, in which case having precise estimates and tight confidence intervals becomes especially important.

same prompting approach as (Li et al., 2024b). We focused on scenarios where only a few of the 16K human preferences were available, and compared our approach (using both available human preferences and all `gpt-4o-mini` preferences) to the classical approach.

Results are shown in Figure 4. We observed that the BT coefficients were better estimated by `PPI++` than by the classical approach, hinting that the point estimates of AutoEval are more accurate (Figure 4a). `PPI++` also produced calibrated, and tighter confidence intervals than the classical approach (Figure 6, Table S5) We also observed ESS showing a 20% to 25% improvement over the classical approach (Figure 4b). Finally, we observed that the estimated rankings of the models were more correlated with the true rankings when using `PPI++` (Figure 4c).

We also studied other choices of LLM judges (Figure 5). Similar to the protein fitness experiment, the quality of the LLM judge had a large impact on the performance of our approach. Furthermore, we observed that all considered judges allowed our approach to outperform the classical approach, obtaining ESS improvements between 20% to 35% depending on the LLM judge.

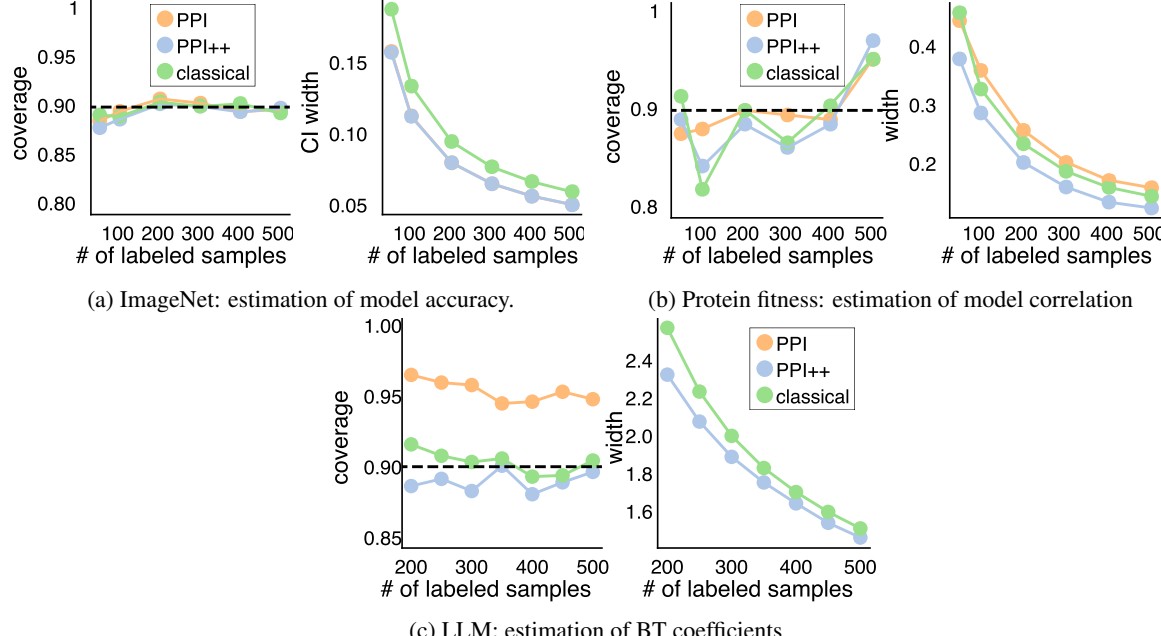

(a) ImageNet: estimation of model accuracy.

(b) Protein fitness: estimation of model correlation

(c) LLM: estimation of BT coefficients

*Figure 6.* **Interval metrics for the different experiments.** Coverage (*left*) and width (*right*) of the 90%-confidence intervals. Each experiment is described in the main text and focuses on the estimation of a different metric.

## Software and Data

All code used to reproduce this work is available as supplementary materials available on OpenReview. We refer the reader to Supplement C for details on the experimental setup. Tools to apply the described methodology for model evaluation are available as a Python package, available at https://github.com/aangelopoulos/ppi_py. Code to reproduce the experiments is available at https://github.com/PierreBoyeau/autoeval.

## Acknowledgements

The authors would like to thank Tijana Zrnić for her feedback and Jessica Dai for helpful conversations. This work was supported in part by the Vannevar Bush Faculty Fellowship program under grant number N00014-21-1-2941.

## Impact Statement

AutoEval provides a principled strategy to facilitate model evaluation in low-data regimes, which is relevant to deploy reliable machine learning systems. AutoEval could also make it easier to certify algorithmic fairness, e.g., by estimating the least-squares coefficients relating sensitive attributes and prediction error. Our methodology could also enable more efficient human oversight of model evaluation, which remains crucial to responsibly deploy machine learning systems.

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

## A. Code snippets

This section provides code snippets to produce confidence intervals and point estimates for model accuracy and pairwise comparisons with the existing Python package `ppi_py` (Angelopoulos et al., 2023a).

```python
from ppi_py import ppi_mean_pointestimate, ppi_mean_ci

# y_labeled <- (n,) ground-truth values of Y_i on labeled dataset
# yhat_labeled <- (n,M) predicted values of Y_i for each model on labeled dataset
# p_labeled <- (n,M) top softmax scores of each model on labeled dataset
# p_unlabeled <- (N,M) top softmax scores of each model on unlabeled dataset
# alpha <- (float) error rate of confidence interval

corrects = (yhat_labeled == y_labeled[:,None]).astype(float)

hat_mu = ppi_mean_pointestimate(corrects, p_labeled, p_unlabeled)
ci_mu = ppi_mean_ci(corrects, p_labeled, p_unlabeled, alpha=alpha)
```

Snippet 1: Python code to produce CIs and point estimates for model accuracy. The variable meanings are explained in the code comments.

```python
from ppi_py import ppi_logistic_pointestimate, ppi_logistic_ci

# X_labeled <- (n,M-1) one row per battle; -1 for model A, 1 for model B, 0 else
# y_labeled <- (n,) 0 if model A wins, 1 if model B wins
# yhat_labeled <- (n,) predicted values of Y_i on labeled dataset
# X_unlabeled <- (N,M-1) one row per battle; -1 for model A, 1 for model B, 0 else
# yhat_unlabeled <- (n,) predicted values of Y_i on unlabeled dataset
# alpha <- (float) error rate of confidence interval

hat_zeta = ppi_logistic_pointestimate(
  X_labeled, y_labeled, yhat_labeled,
  X_unlabeled, yhat_unlabeled
)

ci_zeta = ppi_logistic_ci(
  X_labeled, y_labeled, yhat_labeled,
  X_unlabeled, yhat_unlabeled, alpha=alpha
)
```

Snippet 2: Python code to produce CIs for the Bradley-Terry coefficients (without multiplicity correction). The variable meanings are explained in the code comments. For clarity, the matrix `X_labeled` has one row per pairwise comparison. The $i$th row is a two-hot vector, with $-1$ at position $A_i$ and $+1$ at position $B_i$. The matrix `X_unlabeled` is analogous. Note that `X_labeled` and `X_unlabeled` have only $M - 1$ columns, since $\zeta_1$ does not need to be estimated.

## B. Handling covariate shifts

We here describe alternative AutoEval estimators that can be used to handle covariate shifts. We focus here on a scenario where the labeled inputs $X_i$ are obtained from a different distribution $Q_X$ than the distribution of interest $P_X$. More

particularly, we assume that for any $i \leq n$,

$$
\begin{cases}
X_i \overset{i.i.d.}{\sim} Q_X \\
Y_i \mid X_i \overset{i.i.d.}{\sim} P_{Y|X_i}
\end{cases}
$$

while $X_j^u \overset{i.i.d.}{\sim} P_X$ for $j \leq N$, and $Y_j^u \mid X_j^u \sim P_{Y|X_j^u}$ (though $Y_j^u$ are unobserved). The rest of the assumptions, and notations, are as in the main text.

We assume that the Radon-Nikodym derivative $w := dP_X/dQ_X$ is known, such that an importance sampling approach can be used to correct for the distributional shift.

*Table S1.* Coverage for $\alpha = 0.1$ in the ImageNet experiment under covariate shifts. To introduce covariate shifts, we sampled labeled data points weighted by the probability predicted by ResNet-101 on one of the 1000 ImageNet classes. Importance weights were estimated using self-normalized importance sampling.

| Sample size | Unweighted | Reweighted |
|---|---|---|
| $n = 50$ | 0.5044 | 0.9128 |
| $n = 100$ | 0.3780 | 0.9196 |
| $n = 200$ | 0.2444 | 0.9184 |
| $n = 300$ | 0.1992 | 0.9376 |
| $n = 400$ | 0.1692 | 0.9252 |
| $n = 500$ | 0.1572 | 0.9320 |

Using the same notations as in the main text, the reweighted AutoEval estimator for metric estimation (Equation (4)) becomes:

$$
\hat{\mu}_m^w := \frac{\lambda}{N} \sum_{i=1}^{N} \hat{\mathbb{E}}_{i,m}^u + \frac{1}{n} \sum_{i=1}^{n} \Delta_{i,m}^{\lambda,w}, \tag{7}
$$

where $\Delta_{i,m}^{\lambda} := w(X_i)\phi(f_m(X_i), Y_i) - \lambda \hat{\mathbb{E}}_{i,m}$

Similarly, for pairwise comparisons, the reweighted estimator modifies Equation (6) as

$$
\hat{\zeta} = \underset{\substack{\zeta \in \mathbb{R}^{M-1} \\ \zeta_1 = 0}}{\arg\min} \frac{1}{n} \sum_{i=1}^{n} \left( w(X_i)\ell_\zeta(X_i, Y_i) - \lambda \ell_\zeta(X_i, \hat{Y}_i) \right) + \frac{\lambda}{N} \sum_{i=1}^{N} \ell_\zeta(X_i^u, \hat{Y}_i^u). \tag{8}
$$

Table S1 shows the coverage for the reweighted AutoEval estimator in the ImageNet experiment under covariate shifts. As expected, the original AutoEval estimator does not provide calibrated confidence intervals due to the broken exchangeability assumption between labeled and unlabeled data points. The reweighted AutoEval estimator, on the other hand, provides calibrated confidence intervals.

## C. Experimental details

### C.1. Data acquisition and preprocessing

**ImageNet**   We downloaded model weights from PyTorch's model zoo for the different ResNet models, trained on the training set of ImageNet. We then computed the different models' predictions on the validation set of ImageNet on a high-performance computing cluster.

**Protein fitness**   We relied on ProteinGym[1] to access both the ground-truth fitness values and the predictions of the different protein language models for a specific assay corresponding to IgG-binding domain mutations of protein G (`SPG1_STRSG_Olson_2014`). All fitness scores were normalized as a preprocessing step.

---

[1] https://github.com/OATML-Marks lab/ProteinGym

**LLM**   We considered a subset of the Chatbot Arena dataset aiming to rank twenty recent LLMs (Table S2). The data contained 16K human preferences over pairs of LLM answers to the same (human-provided) prompts. In parallel, using a similar procedure as in (Zheng et al., 2024), we prompted a judge LLM (gpt-4o-mini) to provide its own preferences for the same prompts and LLM answers.

*Table S2.* Overview of the evaluated language models. Models are grouped by their base architecture family/provider.

| Family | Model | Version/Date |
|---|---|---|
| OpenAI | GPT-4 | 2023-03-14 |
| | GPT-4 | 2023-06-13 |
| | GPT-4-Turbo | 2024-04-09 |
| | GPT-4 Preview | 2023-11-06 |
| | GPT-4-Online | 2024-05-13 |
| Anthropic | Claude-3-Opus | 2024-02-29 |
| | Claude-3-Sonnet | 2024-02-29 |
| | Claude-3-Haiku | 2024-03-07 |
| Google | Gemini-1.5-Pro | 2024-05-14 |
| | Gemini-1.5-Flash | 2024-05-14 |
| Meta | LLaMA-3-70B-Instruct | – |
| | LLaMA-3-8B-Instruct | – |
| Mistral AI | Mistral-Large | 2024-02 |
| Other | Command-R | – |
| | Command-R-Plus | – |
| | PHI-3-Medium-4K-Instruct | – |
| | Qwen-1.5-72B-Chat | – |
| | Qwen-2-72B-Instruct | – |
| | Starling-LM-7B-Beta | – |
| | YI-1.5-34B-Chat | – |

## C.2. Methodological details

**Monte Carlo trials**   In all experiments, we randomly split the data into labeled and unlabeled sets 250 times, and computed all point estimates in the main text and in this supplementary material as the average estimate over these splits.

**Model ranking**   To rank models with the different estimators, we computed 90% confidence intervals for the different approaches after Bonferroni correction. Models with overlapping confidence intervals were assigned the same rank.

## C.3. Experimental setup

All AutoEval experiments were run on a workstation with 12th generation Intel (R)Core (TM) i9-12900KF, 128GB of RAM, and on a compute cluster relying on CPU nodes with four cores. We relied on the Python package ppi_py, except for the LLM experiment, for which we relied on Jax to implement PPI and PPI++ for the Bradley-Terry model.

# D. Additional experiments

## D.1. Running times

Table S3 compares the running times for AutoEval with the classical approach for the ImageNet experiment. There, the main bottleneck for AutoEval consisted in obtaining synthetic labels, with prediction time scaling approximately linearly with the number of unlabeled examples. Once these labels are obtained, AutoEval ran extremely fast, in a few milliseconds.

*Table S3.* Runtime comparison between AutoEval and the classical approach for the evaluation of ResNet-101 in the ImageNet experiment ($n = 1,000$, $N = 50,000$). This experiment was run on a workstation with an Nvidia RTX 3090 GPU, 128GB RAM, and an i9-12900KF CPU.

| Method | Prediction (s) | Inference (ms) |
|--------|----------------|----------------|
| Classic | 5 | 0.3 |
| PPI++ | 237 | 8.3 |

## D.2. Larger sample sizes

*Table S4.* ImageNet experiment for $n = 10,000$.

| Method | MSE ($\times 10^{-5}$) | Interval width | Coverage | Efficiency ratio |
|--------|------------------------|----------------|----------|------------------|
| Classic | 1.43 | 1.37 | 0.93 | 1.00 |
| PPI | 1.07 | 1.21 | 0.931 | 1.27 |
| PPI++ | 1.03 | 1.19 | 0.93 | 1.29 |

## D.3. Coverage analysis

*Table S5.* Coverage analysis for different confidence levels in the LLM experiment. Values represent the proportion of confidence intervals that contain the ground truth parameter across different significance levels ($\alpha$).

| Method | $\alpha = 0.05$ | $\alpha = 0.1$ | $\alpha = 0.15$ | $\alpha = 0.2$ |
|--------|-----------------|----------------|-----------------|----------------|
| Classic | 0.96 | 0.90 | 0.85 | 0.80 |
| PPI | 0.97 | 0.92 | 0.88 | 0.82 |
| PPI++ | 0.95 | 0.90 | 0.85 | 0.79 |

