# OpenReview forum: "AutoEval Done Right: Using Synthetic Data for Model Evaluation"
_ICML.cc/2025/Conference — ICML 2025 poster_

### Official Review · Reviewer_f29Q · 2025-02-26

**Overall Recommendation:** 4

**Summary:**

- Paper addresses the problem of evaluating ML models with limited human validation data.
- Proposes Autoeval, an approach pairing limited human data with large amounts of AI synthetically labeled data to get model eval scores
- The primary contribution is the framework based on the existing PPI work
- Evaluations are done to estimate performance metrics directly (e.g. accuracy) and also relative model performance in the case of BT-models
- Results are positive compared to alternative approaches

**Claims And Evidence:**

Well supported claims
 - Autoeval can improve sample efficiency without introducing bias.
- PPI provides more accurate and reliable estimates than classical methods.
- Autoeval extends to ranking approaches as well.

Unsupported
 - Coverage guarantees are not always obtained as shown in the results and depend on the number of labeled samples (even in cases with large CI width)
- This shows perhaps the claim on IID is not met between labeled and unlabelled

**Essential References Not Discussed:**

see above for references

**Experimental Designs Or Analyses:**

- The experimental designs are thorough and well-executed:

- The authors use a diverse set of domains (computer vision, protein fitness, LLM evaluation) to demonstrate broad applicability + good use of baselines

- It would be interesting though to understand where the method works well and where it fails. e.g. where is the approach poor, where is the correlation low (is it specific types of examples). This would greatly help with understanding

- Also it would be nice to have some runtime and cost information vs classical methods — to understand the trade-off of the gains

**Methods And Evaluation Criteria:**

- The selection of benchmark datasets is strong and covers a wide variety of areas
 - Method is well motivated as well as the evaluation criteria

Just flagging an assumption of the annotator model quality: The method assumes that the AI-generated synthetic labels are at least weakly correlated with the true human labels.

That said the correlations are pretty low. Is this a function of LLMs in general and why specifically are other LLMs better than others? i.e. where are they good and where are they bad

**Other Comments Or Suggestions:**

- possible add a section on the distribution shift problem and how this could be handled
- add a section on how this would work for fairness, as it’s mentioned but never shown
- Deconstruct the method more to understand where it succeeds and where it fails — especially why it fails (types of situations)

**Other Strengths And Weaknesses:**

Strengths:
- clear and well-written paper
- strong results empirically and good theoretical analysis

Weaknesses:
- Issues around novelty — seems like a repackaging of PPI for a different use case (i.e. application).
- Computational cost uncertain
- Needs to augment related work to position better as discussed above
- Add discussion of how this work might be extended under distribution shift in greater detail than the current passing references. Since likely in reality this would be use-case.

**Questions For Authors:**

See above points

**Relation To Broader Scientific Literature:**

In general pretty good

- It would be useful to augment the related work to position the work with respect to recent works on:
- LLM judges:  For example: https://arxiv.org/abs/2403.02839, https://arxiv.org/abs/2412.05579
- Synthetic data for model evaluations. For example: https://arxiv.org/abs/2310.16524

**Theoretical Claims:**

- Theoretical claims are sounds and mainly defer back to the original PPI work & its guarantees
- However, Sec 2.2 does do some solid theoretical analysis

---

> ### Author Rebuttal · Authors · 2025-04-01
>
> We thank you for your thoughtful comments and positive feedback on the relevance of our work, the strength of our benchmark, and of our theoretical analyses. Below are detailed answers to your comments.
>
> **Understanding when AutoEval > classic.**
>
> Our approach adjusts to the quality of synthetic labels. In particular, AutoEval can rely on power tuning to identify the optimal λ used in Equation 4 or 6. This mechanism ensures that our estimates (i) have small asymptotic variance (ii) ignore synthetic labels when these are unreliable. Thus, our method is at least as good as the classical approach and typically better with informative synthetic labels.
>
> We also cover several scenarios in our work where multiple synthetic annotation schemes could be competing. Figures 3 and 5 show that the correlation between synthetic and true labels is a good indicator of AutoEval performance and recommend using this metric to prioritize annotation schemes.
>
> In the context of LLMs, we indeed note overall small correlations between human and LLM judge preferences, likely due to noise in human preferences and the ambiguity of many real-world prompts (multiple valid responses may exist). For instance, agreement rates of 85-90% among experts and 75-80% between experts and crowds have been reported in the past (see Chiang et al. 2024).
> These small correlations also highlight the limitations of LLM judges, biased by answer length, position, and stylistic similarities (Zheng et al. 2023).  Mitigating bias via careful prompting, randomization of answer order (Li et al. 2024), and stronger judges (e.g., via fine-tuning on human preferences) could improve effective sample sizes of AutoEval.
>
> You also highlight the relevance of understanding which LLM judges produce high-quality synthetic labels and on what inputs. Future work could apply AutoEval to specialized areas for LLMs where validation data is too scarce to provide insights into underrepresented tasks, providing better insight into what constitutes good LLM judges across domains.
>
> **Covariate shifts**
>
> Covariate shifts are central to key applications of AutoEval. A common scenario arises when unlabeled inputs are drawn from a distribution different from the target distribution. Labeled inputs could themselves not be sampled from the distribution of interest. These effects are relevant to address fairness concerns. Applied to LLM evaluation, for instance, covariate shift toward specific prompts can lead to biased evaluations that are not representative of target use cases.
>
>
> We will add a supplementary note on AutoEval estimators under covariate shifts. In the canonical case, when unlabeled and labeled inputs respectively drawn from distributions P and Q, we can apply AutoEval using traditional covariate shift adjustment methods. Key assumptions here are that P is absolutely continuous with respect to (w.r.t.) Q and that the Radon-Nikodym derivative of P w.r.t. Q is known.
>
> **Novelty**
>
> Our work is more than a repackaging of PPI for model evaluation. We outline specific theoretical and methodological novelties in our response to Reviewer 3tuY and refer you to that discussion for details.
>
> **Runtime**
>
> We compared runtimes between AutoEval and classical in the ImageNet experiment:
>
> Table: Runtime comparison between AutoEval and the classical approach for the evaluation of Resnet101 in the ImageNet experiment ($n=1,000, N=50,000$). This experiment was run on a workstation with an Nvidia RTX 3090 GPU, 128GB RAM, and an i9-12900KF CPU.
> | Method  | prediction (s) | inference (ms) |
> | :------ | :------------- | :------------- |
> | classic | 5 | 0.3 |
> | PPI++   | 237 | 8.3 |
>
> The main bottleneck here is synthetic label generating, typically scaling linearly with the number of samples. This cost is generally acceptable given the gains in evaluation efficiency and statistical power. We will include this analysis in the revised Supplement.
>
>
> **Other**
>
> > Coverage guarantees are not always obtained [...]
>
> Undercoverage in Figure S1b is due to insufficient Monte Carlo (MC) trials. Rerunning the experiment with $K=500$ MC trials yields:
>
> Table: Coverage for $\alpha=0.1$ in the protein fitness experiment
> | Method  | n=50 | n=100 | n=200 | n=300 | n=400 | n=500 |
> | :------ | :--- | :---- | :---- | :---- | :---- | :---- |
> | PPI     | 0.87 | 0.89  | 0.89  | 0.89  | 0.89  | 0.88  |
> | PPI++   | 0.88 | 0.89  | 0.89  | 0.89  | 0.89  | 0.9   |
> | classic | 0.89 | 0.89  | 0.89  | 0.89  | 0.89  | 0.9   |
>
> The coverage is now much closer to nominal levels and validates our coverage claims.
>
> > It would be useful to augment the related work [...] on LLM judges [...] synthetic data for model evaluations.
>
> We thank you for the suggestion. Since LLM judges are core to our work, we will update the related work in the revision with the suggested references.
>
> > I recommend moving figure S1 to the main paper [...]
>
> We thank you for this comment and will move S1 to the main paper in the revised paper.

---

> > ### Comment · Reviewer_f29Q · 2025-04-02
> >
> > Thank you for your detailed response. I appreciate the clarifications on runtime, coverage guarantees (but see comment below) and novelty.
> >
> > I have the following points remaining to discuss:
> >
> > 1. It is important to still have the limitations analysis: concrete empirical study of failure modes — like where exactly AutoEval should not be used such that a reader can have an understanding to better know where the framework works and where it fails.
> >
> > 2. While I appreciate the theoretical discussion around covariate shift. Empirically, it would be useful to see how the framework is affected under shift & exchangeability is violated. Unless the below is the setting
> >
> > 3. On coverage guarantees — if exchangeability is satisfied, shouldn’t the marginal coverage guarantees be satisfied? i.e. 0.9 or greater. Hence, while the results from the table are close why are the guarantees not satisfied? Is it that exchangeability is not satisfied?
> >
> > Looking forward to hearing from the authors

---

> > > ### Author Response · Authors · 2025-04-07
> > >
> > > Thank you for your follow-up comments. We hope our responses address your concerns, and we would welcome any further questions you may have.
> > >
> > > ## Covariate shifts and failure modes
> > >
> > > We agree with your two first points.
> > > Based on your suggestions, we will include a more comprehensive discussion on AutoEval's assumptions, and how these assumptions might or might not be violated in practical settings. Space permitting in the main manuscript, or otherwise in the Supplement referenced in the discussion, we plan to include the following:
> > >
> > > > A key assumption to apply AutoEval is that unlabeled and labeled inputs are i.i.d. draws from the same distribution. This assumption typically applies when labeled inputs are randomly sampled from the unlabeled pool.
> > > > When there are distributional shifts between the labeled and unlabeled inputs, this assumption might be violated.
> > > > These distributional shifts typically arise when the labeled inputs are not sampled uniformly at random from the unlabeled pool, or when labeled and unlabeled data points come from different populations altogether.
> > > > In such cases, AutoEval, in the form described in Equation 4 and 6, loses the statistical guarantees we outline in the paper, and confidence intervals may no longer be valid.
> > > > To address this issue, we derived alternative AutoEval estimators that are robust to covariate shifts (see Supplement).
> > > > As an illustration, we revisited the ImageNet experiment, with a new setting where the labeled and unlabeled data points are not exchangeable (Table B).
> > > > In this setting, we found that the confidence intervals from AutoEval estimator from Equation 4 were dramatically overconfident, most likely because our experiment introduced strong covariate shifts between labeled and unlabeled inputs.
> > > > Our reweighted AutoEval estimator, on the other hand, provided calibrated confidence intervals.
> > > > These results demonstrate that the original formulation of AutoEval is sensitive to covariate shifts.
> > > > A critical requirement for valid application is therefore careful verification of AutoEval's assumptions.
> > > > When exchangeability is violated, alternative strategies, such as our reweighting approach, must be employed to maintain statistical validity.
> > >
> > > Table B: Coverage for $\alpha=0.1$ in the Imagenet experiment under covariate shifts. To introduce covariate shifts, we sampled labeled data points weighted by the probabilty predicted by Resnet101 on one of the 1000 ImageNet classes.
> > >
> > > | AutoEval estimator | unweighted  | reweighted  |
> > > | ------------------ | ----------- | ----------- |
> > > | n=50               | 0.5044      | 0.9128      |
> > > | n=100              |  0.3780     | 0.9196      |
> > > | n=200              | 0.2444      | 0.9184      |
> > > | n=300              | 0.1992      | 0.9376      |
> > > | n=400              | 0.1692      |  0.9252     |
> > > | n=500              | 0.1572      |  0.9320     |
> > >
> > > In addition to this discussion, we plan to include a detailed description of the reweighted AutoEval estimator in the Supplement, implementing the strategy proposed in our last response.
> > >
> > > Over, we believe this discussion better highlights the importance of exchangeability to produce valid inferences, while providing practical guidance on how to proceed when exchangeability is violated.
> > > We thank you for this suggestion, and would be happy to discuss this further.
> > >
> > > ## Coverage guarantees
> > >
> > > Regarding your third point, we would like to clarify that in the data we showed in our last response, exchangeability was satisfied.
> > > In addition, the fact that the empirical coverage is lower than the nominal coverage (e.g., 0.88 instead of 0.9) is not statistically significant.
> > > To validate this claim, we rerun the coverage experiment from our last response, this time with confidence intervals on the coverage estimates (Table C).
> > > As you can see, the 0.9 nominal coverage is contained in all cases, showing that the coverage guarantees hold empirically.
> > >
> > > Table C: Coverage for $\alpha=0.1$ in the Imagenet experiment with 95% asymptotic confidence intervals on the coverage estimates.
> > >
> > > | Method | n=50 | n=100 | n=200 | n=300 | n=400 | n=500 |
> > > |---------|-------|--------|--------|--------|--------|--------|
> > > | PPI | 0.867 $\pm$ 0.096 | 0.886 $\pm$ 0.064 | 0.893 $\pm$ 0.044 | 0.890 $\pm$ 0.036 | 0.890 $\pm$ 0.031 | 0.882 $\pm$ 0.029 |
> > > | PPI++ | 0.881 $\pm$ 0.092 | 0.887 $\pm$ 0.063 | 0.891 $\pm$ 0.044 | 0.886 $\pm$ 0.037 | 0.889 $\pm$ 0.031 | 0.897 $\pm$ 0.027 |
> > > | classic | 0.887 $\pm$ 0.089 | 0.887 $\pm$ 0.063 | 0.895 $\pm$ 0.043 | 0.887 $\pm$ 0.037 | 0.891 $\pm$ 0.031 | 0.898 $\pm$ 0.027 |

---

### Official Review · Reviewer_uXP6 · 2025-03-14

**Overall Recommendation:** 4

**Summary:**

This paper proposes an approach called "autoevaluation". Given a small set of human-labelled examples, and a larger set of (iid) unlabelled examples, the proposed algorithm can synthetically assign labels in a comparatively efficient and unbiased manner. The authors validate their approach using experiments on real-world tasks such as ImageNet, Protein fitness prediction, and chatbot arena.

**Claims And Evidence:**

Yes, the major claims made in this paper seem well-supported.

**Essential References Not Discussed:**

N/A

**Experimental Designs Or Analyses:**

The experiments presented are a straightforward application of the proposed approach and look sound to me.

**Methods And Evaluation Criteria:**

Yes the proposed PPI-based approach appears to be sound. The empirical evaluation is on standard datasets.

**Other Comments Or Suggestions:**

N/A

**Other Strengths And Weaknesses:**

One issue I have with this line of work is that it makes a set of assumptions (1) existence of unlabelled inputs, and (2) the unlabelled inputs are iid. Considering the state of evaluation in ML wrt to large models, I think it has become increasingly important to curate more challenging evaluation benchmarks. In this respect, generating (or labelling) more examples which are iid to benchmarks that we already have (and have presumably been saturated by large models) doesn't really feel like a particularly impactful step forward towards solving our evaluation troubles. Even if we design a new (not-yet-saturated) benchmark with a few human-labelled examples, how valuable is it actually going to be to scale it up with AI-based labelling? Further note that one of the actual bottlenecks in creating effective benchmarks is designing the "hard inputs" (which this paper assumes we already have access to).

**Questions For Authors:**

N/A

**Relation To Broader Scientific Literature:**

There is growing interest in synthetic data generation and annotation in the field of evaluation considering the fact that human annotation is expensive, and sometimes impractical.

**Theoretical Claims:**

Yes the calculation of confidence intervals in section 2.2 looks correct to me. However, I am not an expert in this area so it is possible I may have missed something.

---

> ### Author Rebuttal · Authors · 2025-04-01
>
> Thank you for your kind and insightful comments. We are grateful for your appreciation of our work, particularly our methodologically sound approach and practical demonstration of efficient model evaluation across multiple domains.
>
> Regarding your comment on the relevance of AutoEval to address benchmark saturation, we would like to highlight a central application that motivates our work. One major inspiration for this work is the chatbot arena project (Chiang et al. 2024). This project serves as a dynamic alternative to traditional static benchmarks, allowing human participants to compare and vote on large language model (LLM) responses..
>
> Despite the availability of large amounts of human validation data, resolving ties between models remains challenging in this project.
> First, many users prompt models with questions but decide not to provide preferences between models. Consequently, a large number of conversations without human preferences is available, which cannot be leveraged by traditional model evaluation approaches.
>
> There are also many cases where there is not enough validation data to resolve ties between LLMs. This can happen when a new LLM releases, requiring large amounts of votes (which may correspond to several days of traffic) to quantify its relative performance.
> Unresolved ties also occur when evaluating model performance on specific applications where relevant conversations are limited, such as coding assistance or creative writing, for instance. Helping resolve ties timely in dynamic benchmarks such as Chatbot Arena with AutoEval is one way we believe our work can help address benchmark saturation.
>
> We agree with your observation regarding the i.i.d. assumptions in our work. In the chatbot arena, for instance, these assumptions are challenged when users ask multiple questions or repeat the same question, leading to data dependencies. To address this, future work should extend AutoEval to more general settings with data dependencies to produce reliable inferences.
>
> Finally, AutoEval does not solve the problem of designing hard inputs. It builds on the assumption that inputs are easy to sample but harder to label. While this assumption is limiting, the applications of our work—including in NLP—show that this paradigm can be useful. Beyond NLP, labeling at scale remains a significant challenge in many domains, such as biomedical applications, where we believe AutoEval can make an impact in facilitating model evaluation.

---

### Official Review · Reviewer_3tuY · 2025-03-18

**Overall Recommendation:** 1

**Summary:**

This paper studies an important question in auto-evaluation --- how to efficiently combine model prediction (imputed output) for abundant unlabeled data with limited gold-standard data to obtain efficient estimation for expected metrics for underlying distributions. The paper's method is a direct result from a recent line of paper in prediction-powered inference, the only difference is that they used for evaluation instead of inference. Experiments have been done for interesting applications such as pairwise comparison etc.

**Claims And Evidence:**

Yes, the claims are clear and evidence seems good.

**Essential References Not Discussed:**

NA.

**Experimental Designs Or Analyses:**

I think the experimental designs and fine but still limited. For instance, coverage did not include for most of the experiments, and whether other metrics beyond MSE could be considered is not discussed. For instance, some cross entropy type of metrics are very widely used, but not discussed as metrics here. I understand this work largely studies unbiased estimator, so if the final aim is MSE, then the only thing we need to do is to reduce variance. But I would like to see other metrics discussed.

**Methods And Evaluation Criteria:**

Yes, it makes sense. Mostly about directly measuring MSE and effective sample size. But this method could also provide confidence intervals based on asymptotic normality, and this paper did not include that in most of experiments, only in one in Appendix C.

**Other Comments Or Suggestions:**

Please see above.

**Other Strengths And Weaknesses:**

I think the problem itself is important but theoretical innovation is quite limited. Also, the experimental results are not comprehensive enough (see above comments).

**Questions For Authors:**

NA.

**Relation To Broader Scientific Literature:**

I think it is useful in practice, but also am concerned that the results by simply using gold-standard data are already good enough, the improvement seems marginal.

**Theoretical Claims:**

The theoretical claims are correct but largely could be directly derived from PPI++, did not have much innovation. I wouldn't claim any innovation in theory for this paper.

---

> ### Author Rebuttal · Authors · 2025-04-01
>
> We thank you for your detailed and thorough feedback. We are grateful for your acknowledgement of the importance of the problem we are tackling, on the clarity of our claims and the methodological soundness of our approach.
> Your review raises a number of points relating to the theoretical contributions and to the experimental design of our work. Please find below our detailed response to these points, that we hope will address your concerns.
>
> **Lack of theoretical contributions.**
>
> We would like to address your concerns about the theoretical contributions of our work. While our approach indeed builds upon the prediction-powered inference (PPI) framework, it contains several novel theoretical and modeling aspects, relevant both in the context of auto-evaluation and PPI.
>
>
> The first contribution is to build a general framework for model evaluation using synthetic labels.
> For metric-based evaluation, our framework accommodates various and nontrivial synthetic annotation schemes (e.g., external model annotation, or self-annotation, see Section 2.1), and allows both for hard or soft synthetic annotations. We also extend this framework to comparison-based evaluation, enabling rigorous model performance inference from pairwise comparisons.
> Our work also introduces chi-squared confidence sets for PPI, an important extension of the original PPI machinery that is not covered in the original papers. Indeed, PPI and its extensions typically focus on simple, low-dimensional inference tasks.  The realization that chi-squared confidence sets can be used may open up new research directions to extend the PPI framework to higher-dimensional problems.
> These developments, along with the demonstration of the practical applicability of our framework across a variety of tasks and domains, highlight that our work is more than a direct result of the PPI framework.
>
> More importantly, we believe our work is particularly timely and of significant scientific relevance given the growing need for efficient evaluation of increasingly complex machine learning models across scientific disciplines. This need is particularly obvious in the context of LLMs, where even crowdsourced validation like Chatbot Arena (Chiang et al. 2024), studied in our work, often suffers from insufficient data to resolve ties between models.
>
>
> **Marginal improvements relative to the classical approach**
>
> The other concern we would like to address relates to the perceived magnitude of improvement over the classical approach. In our experiments, AutoEval reaches efficiency ratios (effective sample sizes over sample sizes) between 1.25-1.40.  Another way to appreciate these numbers is to think about the width of the obtained confidence intervals. An efficiency ratio of 1.25 translates into a 11% reduction in the width of confidence intervals, which, as shown in our experiments, significantly facilitates model ranking. See for instance Fig. 1c, where tighter confidence intervals translate for most sample sizes into at least a two-fold improvement in predicted rank correlation with ground-truth.
>
> We believe these numbers are meaningful and highlight the practical relevance of our work in a variety of ML settings. That being said, the practical relevance of this improvement depends on the financial and time costs to produce human and synthetic validation data. Our method is particularly relevant in instances where human validation cannot be obtained in a timely manner. Examples include the Chatbot arena, where human annotation depends on web traffic, which cannot be controlled, or in biological assays where obtaining validation data may take weeks or months, but where producing large amounts of synthetic data can be done in a few hours.
>
>
> **Benchmark**
>
> > But this method could also provide confidence intervals based on asymptotic normality, and this paper did not include that in most of experiments [...].
>
> We would like to make an important clarification, as building valid confidence intervals of model performance is central to our work. In particular, Figure S1 of Appendix C shows confidence intervals not for one, but all of the main experiments (in Sections 2.3, 2.4, and 3.3) considered in this work.
>
> These confidence intervals, which we demonstrate are well-calibrated and tighter than classical approaches in all experiments, are fundamental to our methodology as they enable statistically rigorous model comparison and selection even with limited labeled data.
>
>
> > I would like to see other metrics discussed.
>
> We thank you for suggesting the exploration of additional metrics. In response, we have extended our analysis in the following way. We studied confidence interval coverage for multiple alpha values (0.8, 0.85, 0.9, and 0.95) beyond the original α=0.1, confirming consistent calibration across confidence levels.
> Regarding cross-entropy metrics, we would appreciate your clarification on how you envision these metrics being applied in our evaluation framework.

---

> > ### Comment · Reviewer_3tuY · 2025-04-09
> >
> > Thanks for the response.
> >
> > 1. I think the claimed contribution in theory is not actually considered theoretical contribution, I would rather consider it is just different specific application scenarios. The key core theoretical tool is just PPI++, with an even simpler form, it will not need to go into Taylor expansion for M-estimation etc. It is just a simple application of CLT for i.i.d random variable.
> >
> > 2. I take a look at Appendix C, I don't know whether it is the demonstration problem at my end, I did not see the results of intervals. Actually, Figure a and c in Appendix C are not showing up at all!
> >
> > I will remain my score.

---

> > > ### Author Response · Authors · 2025-04-09
> > >
> > > Thank you for your follow-up. Regarding point 1, we respect your assessment of our theoretical contributions while maintaining our view,
> > > outlined in our previous response, that the work offers meaningful extensions to the PPI framework in the context of evaluation.
> > >
> > > For point 2, we apologize for the technical issue you experienced with Figure S1 in Appendix C.
> > > We confirm, after manual inspection, that Figures S1a and S1b are included in our submission: https://openreview.net/pdf?id=S8kbmk12Oo
> > > On our end, these figures are properly displayed on OpenReview using a Chrome or Firefox browser and on Adobe Acrobat Reader after manual download.
> > >
> > > To reproduce your issue, however, we did observe that figures S1a and S1c are not properly displayed on Safari.
> > > We will make sure to understand this issue and fix it in future versions of the manuscript.
> > > Meanwhile, we invite you to download the pdf or use another browser in case you are using Safari.

---

### Official Review · Reviewer_GnRr · 2025-03-21

**Overall Recommendation:** 2

**Summary:**

The goal of this work is to reduce the cost and time of evaluating machine learning models using AI-labeled synthetic data. Introduces algorithms for auto evaluation that improve sample efficiency while remaining unbiased.

**Claims And Evidence:**

•	This problem has been tackled in the literature with different names: pseudo-labeling, curriculum learning, consistency regularization etc. It is important to compare the proposed method against previously published work.

•	Evaluation of LLM responses from pair-wise responses has also been studied in measuring uncertainty using structural similarity and other pair-wise metrics.

•	The proposed problem statement is relevant for training samples as well.

•	The effective sample sizes considered are very small, and important to evaluate the approach with larger datasets.

•	The claim that proposed methodology provides calibrated and tight confidence intervals is clearly apparent (presented in supplemental material). Need to compare with other principled uncertainty metrics.

**Essential References Not Discussed:**

-

**Experimental Designs Or Analyses:**

see above sections.

**Methods And Evaluation Criteria:**

The approach is evaluated on Imagenet, protein fitness experiment and pairwise preferences for LLMs. The evaluation is limited.

**Other Comments Or Suggestions:**

-

**Other Strengths And Weaknesses:**

-

**Questions For Authors:**

-

**Relation To Broader Scientific Literature:**

-

**Theoretical Claims:**

There is no clear theoretical justification for PPI++ metric.

---

> ### Author Rebuttal · Authors · 2025-04-01
>
> We thank you for your comments. We appreciate your recognition of our work's practical relevance in reducing ML evaluation costs and the strength of our benchmark.
>
> Please find below point-by-point answers to your comments.
>
> **Positioning of our approach relative to semi-supervised training strategies.**
>
> > This problem has been tackled in the literature with different names[...]. It is important to compare [...] against previously published work.
>
> Thank you for the comment. Although these previously published topics handle related topics, our work is very different and does not fit in these existing areas. The fundamental reason is simple: our work is about evaluation, and your suggested topics are about model training.
>
> To elaborate, pseudo-labeling uses predicted labels for unlabeled data as ground-truth during training (Lee 2013; Arazo et al. 2020).  Curriculum learning is another model training paradigm that progressively increases training difficulty, e.g., via multi-stage training, to improve generalization (Bengio et al. 2009). Consistency regularization aims to enforce semantically similar inputs to have similar label predictions, relevant in semi-supervised training settings (Bachman, Alsharif, and Precup 2014; Laine and Aila 2016).
>
> The key difference is that our focus is on the reliable evaluation of already trained models, rather than the training of these models on limited data.  Our framework provides strong guarantees on the behavior of model performance estimates which is not the case for the mentioned literature. While these guarantees may not be relevant in a model training setting, they become crucial when evaluating models before deployment.
>
> > The proposed problem statement is relevant for training samples as well.
>
> While PPI can be employed for training purposes, this direction is beyond our scope, and we make no claims for training applications. We plan to revise our related work to cover synthetic label for model training to emphasize distinctions with our work.
>
> > Evaluation of LLM responses [...] has also been studied in measuring uncertainty using structural similarity and other pair-wise metrics.
>
> Thank you for pointing towards this related research area. A growing body of literature has indeed focused on evaluating LLMs from pairwise comparisons, often using LLMs as judges. We will revise the manuscript to include a more comprehensive and detailed discussion of this literature by including the references mentioned by Reviewer f29Q.
>
>
> **Benchmark**
>
> > The effective sample sizes considered are very small, and important to evaluate the approach with larger datasets.
>
> The labeled sample sizes in our work are small to illustrate practical settings with limited human validation; however, the results certainly hold for larger sample sizes.  To address this point head-on, we studied the behavior of AutoEval for larger sample sizes in the imagenet experiment, as shown below:
>
> Table: ImageNet experiment for $n=10,000$.
> | Method | MSE (1e-5) | Interval width | Coverage ($\alpha = 0.1$) | Efficiency ratio ($ESS / n$) |
> |---------|-----------|----------------|-----------|-----------------|
> | classic | 1.43 | 1.37 | 0.93 | 1.00 |
> | PPI | 1.07 | 1.21 | 0.931 | 1.27 |
> | PPI++ | 1.03 | 1.19 | 0.93 | 1.29 |
>
> Above, AutoEval compares favorably to the classical approach for pointwise performance evaluation. It also provides tighter confidence intervals, large effective sample sizes, and proper coverage.
>
> > The claim that proposed methodology provides calibrated and tight confidence intervals is clearly apparent [...]. Need to compare with other principled uncertainty metrics.
>
> To answer your comment, we conducted additional analyses to evaluate confidence interval calibration across significance levels (0.8, 0.85, 0.9, and 0.95), and compared directly with the classical approach. The results, to be included in the revised Supplement, demonstrate our confidence intervals maintain proper calibration across these significance levels.
> Empirical coverage closely matches the nominal coverage, indicating well-calibrated uncertainty estimates for AutoEval. We hope this addresses your concern, and would be happy to implement other metrics if you have other suggestions.
>
> > The approach is evaluated on Imagenet, protein fitness experiment and pairwise preferences for LLMs. The evaluation is limited.
>
> The current manuscript describes several canonical applications of our framework, for a variety of tasks, in model evaluation and ranking across different domains. We believe that the current benchmark offers a comprehensive evaluation of the proposed framework in a variety of real-world settings.
>
>
> **Other comments**
>
>
> > “There is no clear theoretical justification for PPI++ metric.
>
> We would be happy to take your suggestions into account to clarify the distinctions between PPI and PPI++ if our description lacked clarity. Could you expand on which specific section of the paper you are referring to?

---

### Decision · Program_Chairs · 2025-05-01

**Decision:**

Accept (poster)

**Comment:**

This paper proposes an evaluation method that incorporates synthetic labels along with human-labeled test data to obtain better estimate of a given model’s accuracy. The proposed method uses PPI++ to use synthetic labeled data. The proposed method is simple and applicable to various types of supervised learning settings. The efficacy of the proposed method is demonstrated by evaluation on various scenarios, including classification tasks (image recognition), regression (protein fitness prediction) and rankings (pairwise comparisons for chatbot arena). The theoretical innovations are limited. Another limitation of the proposed method as noted by reviewers is the assumption of i.i.d. data. The proposed re-weighting of the evaluation metric requires knowledge of the distribution. That said, sample-efficient evaluation of models is a very challenging task and the proposal of  applying PPI++ for model evaluation incorporating synthetic labels is a promising direction. Overall, the strengths of the submission outweigh the limitations.

Please include the additional experimental results and discussions on covariate shifts and limitations of the current approach that came up during review-rebuttal period.